# Does knowledge of pregnancy complications influence health facility delivery? Analysis of 2014 Bangladesh Demographic and Health Survey

**Edward Kwabena Ameyaw**[1]ᵒ, **Bright Opoku Ahinkorah**[1]ᵒ, **Abdul-Aziz Seidu**[2,3]ᵒ *

**1** School of Public Health, Faculty of Health, University of Technology Sydney, Sydney, NSW, Australia,
**2** Department of Population and Health, College of Humanities and Legal Studies, University of Cape Coast, Cape Coast, Ghana, **3** College of Public Health, Medical and Veterinary Sciences, James Cook University, Townsville, QLD, Australia

ᵒ These authors contributed equally to this work.
* abdul-aziz.seidu@stu.ucc.edu.gh

## Abstract

### Introduction

Only thirty-seven percent (37%) of deliveries occur in health facilities in Bangladesh despite the enormous benefits of health facility delivery. We investigated women's recall of receiving counseling on pregnancy complications and how it affects health facility delivery in Bangladesh.

### Materials and methods

Data from the 2014 Bangladesh Demographic and Health Survey was used for the study. After calculating the proportion of women who were informed about pregnancy complications during their last Antenatal Care (ANC) and the number of them who delivered in health facilities, Binary Logistic Regression was utilized in investigating chances of giving birth in health facilities among women who recalled they were told about pregnancy complications and those who were not told. The models were considered significant at 95%.

### Results

A little above half of the women who were told about pregnancy complications during ANC delivered in health facilities (53.3%) and 43.6% of those who were not told delivered in health facilities. The findings revealed that women who were told about pregnancy complications during ANC were more likely to deliver at the health facility compared to those who were not told [COR = 1.56, CI = 1.31–1.87], and this persisted after controlling for the effect of covariates [AOR = 1.44, CI = 1.21–1.71].

### Conclusion

This study has stressed the importance of telling women about pregnancy complications during ANC by revealing that telling women about pregnancy complications during ANC is

**Data Availability Statement:** The data can be accessed freely from the Measure DHS website: https://www.dhsprogram.com/data/dataset/Bangladesh_Standard-DHS_2014.cfm?flag=0.

**Funding:** The author(s) received no specific funding for this work.

**Competing interests:** The authors have declared that no competing interests exist.

likely to result in health facility delivery. Health workers should intensify health education on pregnancy complications during ANC and motivate women to deliver in health facilities.

## Introduction

Globally, maternal mortality ratio (MMR) declined by 44% from 385 to 216 maternal deaths per 100,000 live births between 1990 and 2015 [1]. Meanwhile, the reduction has been unbalanced, with low and middle-income countries (LMICs) experiencing the largest burden. In 2015, the MMR in LMICs was 239 per 100 000 live births while high-income countries recorded 12 per 100 000 live births [2]. The persistently high rates of MMR in LMICs has been attributed to high fertility, high maternal mortality tendency, fragile health structures and poverty [3, 4]. MMR in Bangladesh as of 2015 was 176 per 100 000 live births [1]. Maternal mortality in Bangladesh is linked to imbalances in primary healthcare access with a huge gap between women in advantaged and disadvantaged socioeconomic communities [5]. This is marked by the 15% and 70% health facility deliveries among women in lowest and highest wealth quintiles respectively [6].

In a lot of instances, maternal deaths occur because of direct obstetric conditions including obstructed labor, hemorrhage, unsafe abortion, and hypertensive disorders [7]. Most of the reported complications take place during delivery and some of them are unpredictable. Meanwhile, the complications could be solved effectively and deaths prevented by delivering in health facilities that have skilled birth attendants (SBAs) [8]. The importance of SBAs during delivery was deepened by the WHO's recommendation that every delivery should be overseen by SBA—a health care provider who can recognize and handle normal labor and birth, figure out and provide basic care and referral [9]. However, the global proportion of SBA assisted deliveries is below the recommended levels [10]. In LMICs, SBAs assist in about half of all deliveries and even where ANC is high, most deliveries occur at home [11].

Despite the argument concerning effective ways by which maternal mortality can be reduced in LMICs, programmes are planned with the intent that complications are unpredictable, and prompt professional intervention is required anytime they occur [12]. However, studies have shown that knowledge of obstetric danger signs and preparedness for emergency complications are effective ways of increasing maternal health service utilisation especially seeking skilled care during delivery [13–15]. The new Antenatal Care (ANC) model by WHO aims at providing relevant and timely information that can lead to positive delivery experience [16]. Information about pregnancy complications can assist pregnant women to promptly identify these danger signs and positively influence their subsequent healthcare-seeking actions, including health facility delivery [17, 18]. These positive health utilisation attitudes can be achieved if women are well informed and understand that delivery at home can upsurge complications [13, 19].

The most recent Demographic and Health Survey reported that 37% of deliveries occurred in health facilities [6]. Several studies conducted in Bangladesh have identified some factors that determine health facility delivery. Reported significant factors for health facility delivery include place of residence [20–22], maternal age [21], region of residence [22, 23], educational status of the mother [20–22], parity [22, 24, 25], exposure to media [26] and access to ANC [27, 28].

Although these factors play significant role in a pregnant woman's decision to deliver at the health facility, there are other critical factors such as obtaining information on pregnancy complications during ANC. Notwithstanding, it appears none of the studies on health facility

delivery in Bangladesh has looked at information women obtain about pregnancy complications at ANC and how it affects health facility delivery. We, therefore, hypothesize that women in Bangladesh who obtain information on pregnancy complications at ANC are probable to deliver at the health facility. Findings from the study can enhance information sharing on pregnancy complications in Bangladesh and will stimulate healthcare providers to consistently discuss pregnancy complications with women during ANC and use educational interventions during ANC to encourage women to deliver at the health facility.

## Materials and methods

### Data source

Data underlying our study were obtained from the 2014 version of Bangladesh Demographic and Health Survey (BDHS). This nationally representative survey was conducted under the guidance of the National Institute of Population Research and Training (NIPORT) [6]. It was implemented between June and November 2014 and technical assistance was offered by ICF International through the DHS Program. This is the seventh round of the BDHS which gathered information for addressing the Health, Population and Nutrition Sector Development Program (HPNSDP). It aimed at offering information to managers and policymakers to help them plan and implement interventions successfully. The survey collates evidence on some maternal and child health characteristics such as fertility preferences and utilisation of maternal health services. The survey was done with a two-stage stratified sampling approach based on the 2011 Population and Housing Census of Bangladesh. In the maiden stage, 600 enumeration areas (EAs) were identified through probability proportional to the size of EAs. This resulted in 207 from urban and 393 from rural locations [6]. Household listing was conducted in each EA to develop a sampling frame for the second stage. A sample of 30 households were selected averagely per EA in order to have a statistically dependable estimation of the core health variables for Bangladesh. This resulted in 17,989 households selected for the survey of which 17,300 were successfully interviewed. In those households, 18,245 women eligible for the individual interview were identified, of which 17,863 were interviewed (98% response rate). Details of the survey design, sampling techniques, questionnaire and quality control are narrated in the BDHS 2014 report [6]. We relied on the "Strengthening the Reporting of Observational Studies in Epidemiology" (STROBE) statement writing the manuscript.

### Definition of variables

**Dependent variable.** The dependent variable in this study was place of delivery. During the survey, women were asked "Where did you give birth?" with eleven responses broadly on various health facilities and the respondent's home. To derive the dependent variable, we re-categorised these into a binary outcome where respondent's home was recoded as zero (0) whilst all health facilities were recoded as one (1). Our categorisation into home or health facility followed conceptualisation by some earlier studies that explored place of delivery using the Demographic and Health Survey data of other countries [29–31].

**Independent variables and sample.** The principal independent variable was whether a woman recalls being told about pregnancy complications at the time of her last ANC visit. Only women who had at least one ANC visit with any provider were asked about counseling on danger signs during pregnancy. Therefore, women who had at least one live birth in the 3 years before the survey answered this question. In terms of ANC visit, 964 women who had live births in the past 3 years preceding the survey had never visited ANC and 5 (0.1%) indicated don't know. These women were excluded. In relation to the women who attended ANC and were asked about their recall of counseling on danger signs during pregnancy, the

responses were "No" coded as 0, "Yes" coded as 1 and "Don't Know" coded as 8. Only four (4) women (0.11%) indicated "Don't Know" and they were excluded from the sample since they were unsure whether they were told or not.

As a result, 3,519 women were included in our analysis. To decipher the robustness of our key explanatory variable in relation to health facility delivery, some theoretically pertinent variables were incorporated in the analysis. These are age, education, wealth quintile, place of residence, division, religion, parity and wantedness of the last child. Parity was recoded into one (1), two (2) and three or more. Age was recoded into 15–19, 20–24, 25–29, 30–34, 35–39 and 40+ and religion was recoded into Islam and other.

**Analytical procedure.** We initiated our analysis by calculating the proportion of women who gave birth at home or health facility against whether they were told about pregnancy complications. We also calculated proportions of place of delivery (either home or health facility) by the socio-demographic characteristics of the women (see Table 1). Our inferential analysis comprised two Binary logistic regression models. The first model involved place of delivery and whether a woman was told about pregnancy complications. The outcome of this was reported as Crude Odds Ratio (COR) at a 5% margin of error. In the second model, we included the women's socio-demographic characteristics in order to ascertain how they will be associated with whether a woman was told about pregnancy complications to determine where the women will deliver. The sample was weighted and all analyses were carried out with STATA Version 13.0.

## Ethics

We analyzed secondary data provided by Bangladesh Demographic and Health Survey (DHS) -2014. Ethical clearance for the BDHS 2014 data collection project was obtained from the NIPORT and ICF International's Institutional Review Boards (IRBs). The survey ensured international ethical standards of confidentiality, anonymity, and informed consent. This study is based on publicly available, de-identified DHS data; https://dhsprogram.com/what-we-do/survey/survey-display-441.cfm. The dataset was requested and a letter of data authorization was received from the DHS Program ICF International. More details regarding ethical standards of the DHS data are available at: https://www.dhsprogram.com/What-We-Do/Protecting-the-Privacy-of-DHS-Survey-Respondents.cfm.

## Results

### Descriptive results

As shown in Table 1, 53.3% of women who recall they had information on pregnancy complications during ANC delivered at the health facility compared to 43.6% who had no information on pregnancy complications during ANC. Similarly, health facility delivery was high among women aged 35–39 (51.2%), those with higher education (79.0%), richest women (75.6%), women in urban areas (63.7%), women in Khulna (61.7%), Christians (85.7%), women with parity one (55.7%) and women who wanted their last child at the time of pregnancy (49.8%).

### Binary logistic regression results

The results in Table 2 show that women who were told about pregnancy complications during ANC were more likely to deliver at the health facility compared to those who were not told [COR = 1.57, CI = 1.32–1.87], and this persisted after controlling for the effect of covariates [AOR = 1.45, CI = 1.22–1.73]. The results further showed that the likelihood of health facility

**Table 1. Told about pregnancy complications, socio-demographic characteristics and place of delivery (N = 3,519).**

| Variable | Weighted N | Percentage (%) | Place of delivery | |
|---|---|---|---|---|
| | | | Home % | Health facility % |
| **Told about pregnancy complications ((X$^2$ = 32.6,p<0.001)** | | | | |
| No | 1,875 | 53.3 | 56.4 | 43.6 |
| Yes | 1,644 | 46.7 | 46.7 | 53.3 |
| **Age (X$^2$ = 6.7,p = 0.248)** | | | | |
| 15–19 | 761 | 21.6 | 55.3 | 44.7 |
| 20–24 | 1,187 | 33.7 | 50.9 | 49.0 |
| 25–29 | 899 | 25.5 | 50.0 | 50.0 |
| 30–34 | 470 | 13.4 | 52.7 | 47.3 |
| 35–39 | 163 | 4.7 | 48.7 | 51.2 |
| 40–49 | 39 | 1.1 | 57.9 | 42.1 |
| **Education (X$^2$ = 318.2,p<0.001)** | | | | |
| No Education | 361 | 10.3 | 73.5 | 26.4 |
| Primary | 870 | 24.7 | 65.2 | 34.8 |
| Secondary | 1,851 | 52.6 | 50.1 | 49.9 |
| Higher | 437 | 12.4 | 21.0 | 79.0 |
| **Wealth Quintile (X$^2$ = 499.0,p<0.001)** | | | | |
| Poorest | 554 | 15.7 | 78.5 | 21.5 |
| Poorer | 595 | 16.9 | 68.8 | 31.2 |
| Middle | 696 | 19.8 | 56.9 | 43.0 |
| Richer | 830 | 23.5 | 46.5 | 53.5 |
| Richest | 844 | 24.1 | 24.4 | 675.6 |
| **Place of residence (X$^2$ = 193.8,p<0.001)** | | | | |
| Urban | 1,049 | 29.8 | 36.3 | 63.7 |
| Rural | 2,470 | 70.2 | 60.7 | 39.2 |
| **Division (X$^2$ = 78.8,p<0.001)** | | | | |
| Barisal | 190 | 5.4 | 60.3 | 39.7 |
| Chittagong | 731 | 20.8 | 54.2 | 45.8 |
| Dhaka | 1,329 | 37.8 | 48.7 | 51.3 |
| Khulna | 317 | 9.0 | 38.3 | 61.7 |
| Rajshahi | 339 | 9.7 | 45.3 | 54.7 |
| Rangpur | 354 | 10.0 | 55.7 | 44.3 |
| Sylhet | 259 | 7.3 | 62.2 | 37.8 |
| **Religion (X$^2$ = 6.5,p<0.05)** | | | | |
| Islam | 3,236 | 91.9 | 52.5 | 47.5 |
| Other* | 283 | 8.1 | 44.5 | 55.5 |
| **Parity(X$^2$ = 97.6,p<0.001)** | | | | |
| One | 1,515 | 43.1 | 44.3 | 55.7 |
| Two | 1,086 | 30.8 | 51.5 | 48.5 |
| More than two | 918 | 26.1 | 65.0 | 34.9 |
| **Wanted Last Child (X$^2$ = 24.5,p<0.001)** | | | | |
| Wanted then | 2,661 | 75.6 | 50.2 | 149.8 |
| Wanted later | 512 | 14.6 | 52.6 | 47.4 |
| Wanted no more | 346 | 9.8 | 64.8 | 35.2 |

Source: 2014 BDHS Other Religion* (Hinduism, Buddhism and Christianity).

**Table 2. Effects of knowledge on pregnancy complication and socio-demographic characteristics on health facility delivery.**

| Variable | Model I | | Model II | |
|---|---|---|---|---|
| | COR 95% CI | | AOR 95% CI | |
| *Told about pregnancy complications* | | | | |
| No | 1 | [1,1] | 1 | [1,1] |
| Yes | 1.57*** | [1.32–1.87] | 1.45*** | [1.22–1.73] |
| *Age* | | | | |
| 15–19 | | | 1 | [1,1] |
| 20–24 | | | 1.50** | [1.14–1.96] |
| 25–29 | | | 2.08*** | [1.46–2.96] |
| 30–34 | | | 2.99** | [1.52–5.91] |
| 35–39 | | | 3.60*** | [2.20–5.90] |
| 40–49 | | | 2.34 | [0.74–7.37] |
| *Education* | | | | |
| No Education | | | 1 | [1,1] |
| Primary | | | 1.76** | [1.18–2.61] |
| Secondary | | | 2.31*** | [1.56–3.42] |
| Higher | | | 4.20*** | [2.61–6.79] |
| *Wealth quintile* | | | | |
| Poorest | | | 1 | [1,1] |
| Poorer | | | 1.36* | [1.01–1.84] |
| Middle | | | 1.74* | [1.14–2.65] |
| Richer | | | 2.45*** | [1.77–3.38] |
| Richest | | | 4.78*** | [3.26–7.00] |
| *Place of residence* | | | | |
| Urban | | | 1 | [1,1] |
| Rural | | | 0.65** | [0.50–0.84] |
| *Division* | | | | |
| Barisal | | | 1 | [1,1] |
| Chittagong | | | 1.16 | [0.79–1.70] |
| Dhaka | | | 1.31 | [0.94–1.85] |
| Khulna | | | 2.60*** | [1.77–3.83] |
| Rajshahi | | | 1.85** | [1.29–2.63] |
| Rangpur | | | 1.31 | [0.83–2.05] |
| Sylhet | | | 0.93 | [0.67–1.30] |
| *Religion* | | | | |
| Islam | | | 1 | [1,1] |
| Other* | | | 1.40 | [0.90–2.18] |
| *Parity* | | | | |
| One | | | 1 | [1,1] |
| Two | | | 0.58*** | [0.45–0.76] |
| More than two | | | 0.35*** | [0.25–0.49] |
| *Wanted Last Child* | | | | |
| Wanted then | | | 1 | [1,1] |
| Wanted later | | | 1.17 | [0.89–1.53] |
| Wanted no more | | | 1.03 | [0.74–1.44] |

COR = Crude Odds Ratio, AOR = Adjusted Odds Ratio, CI = Confidence Intervals, Other Religion* (Hinduism, Buddhism and Christianity)

* $p < 0.05$

** $p < 0.01$

*** $p < 0.001$

Source: 2014 BDHS

delivery increases with age, with women aged 40 and above years having the highest likelihood of health facility delivery [AOR = 3.60, CI = 2.20–5.90].

The odds of health facility delivery was also highest among women with higher educational level [AOR = 4.20, CI = 2.61–6.79], richest women [AOR = 4.78, CI = 3.26–7.00] and women living in Khulna [AOR = 2.60, CI = 1.77–3.83]. On the other hand, the likelihood of health facility delivery was low among women in rural areas [AOR = 0.65, CI = 0.50–0.84] and women with more than two children [AOR = 0.35, CI = 0.25–0.49].

## Discussion

This study sought to test the hypothesis that women in Bangladesh who recall they were told about pregnancy complications at ANC are more likely to deliver at the health facility. The inferential analysis confirmed the hypothesis that women who recall getting information on pregnancy complications during ANC have higher odds of delivering in a health facility compared to those who did not receive such information during ANC. The association between getting information on pregnancy complications during ANC and health facility delivery was significant although slightly attenuated in the multivariable analyses when we adjusted for woman's age, educational level, wealth, parity, religion, and region of residence.

This finding affirms several other studies on the relationship between exposure to health information and health behavior uptake. For instance, the study by Dutamo, Assefa, and Egata [32] revealed that women who had knowledge of at least one pregnancy danger sign were more likely to deliver in a health facility. Relatedly, Mageda and Mmbaga [33] found in Tanzania that women who had advice from health care providers to deliver at a health care facility had higher odds to deliver in a health facility. The results corroborate other findings in Ethiopia [34, 35], Eritrea [10], Zambia and Sri Lanka [36], which found that the information women receive during ANC has a positive association with their use of institutional delivery services. As explained by Eshete, Legesse, and Ayana [37], mothers who are exposed to information on pregnancy complications have a greater fear and this can propel them to deliver in a health facility. This is also related to two constructs of the Health Belief Model- perceived severity and cues to action, which state that individuals are more likely to take up positive health behavior when they perceive a condition to be severe [38]. In this sense, pregnant women who get information on pregnancy complications might be afraid of the complications they are likely to face when they deliver at home. Moreover, if women are knowledgeable about danger signs they will be motivated to deliver in health facility to avert possible birth complications.

Situating these findings into the existing program context in Bangladesh, there are several interventions to promote counseling on danger signs during ANC including behavior change communication (BCC), posters and counseling guides. For example in relation to BCC strategies, they are considered to be an integral part of the ways by which pregnant women can acquire knowledge in Bangladesh. These strategies range from individual face-to-face contacts to the use of traditional media like folk songs and street theatre [39]. Specifically, programs such as improving maternal neonatal and child survival (IMNCS) programme of BRAC was initiated to reduce maternal, neonatal and child mortality and morbidity in poor communities [40, 41]. It started in Nilphamari district in northern Bangladesh and was gradually expanded to three other districts in 2008 and a further six in 2010. While the BRAC IMNCS programme mainly targets pregnant women, mothers of newborns and under-5 children, it also includes family members and influential community people in its target population [39]. Messages delivered by Community Health Workers include: pregnancy care, birth preparedness, safe delivery, postpartum care, neonatal and child health and include topics such as: nutrition, safety, rest taking and cleanliness during pregnancy, the need for antenatal check-up.

Importantly, the danger signs of pregnancy, delivery and the postpartum period are discussed during these programs [39–42].

Again in relation to posters the BRAC IMNCS had four different types of posters and two stickers were distributed to the women as part of the BCC intervention during the roll out of the project. The Posters illustrated first, messages such as a healthy mother and child with the message "*if you find any problem in pregnancy, do not delay. We want healthy mothers, healthy children, and healthy neonates*"; One sticker depicts the maternal danger signs of pregnancy, delivery, and post delivery period'. The danger signs include high fever; severe headache and blurring of vision; prolonged labor; convulsion; and hemorrhage or excessive bleeding. The second sticker is a smaller version of the poster illustrating neonatal danger signs. Both stickers and the poster on children's danger signs also highlight procedures for accessing health facilities [39].

Posters and stickers are posted on the wall of the woman's house by the Community Health Workers to reinforce knowledge and awareness. The Program organizer's mobile phone numbers are listed at the bottom of each poster and sticker so that they can be contacted in case of emergencies. According to Rahman et al. [39, 40], the mass media approach to MNCH BCC includes folk songs (locally termed as *jaarigan*) and street theatre (*naatak*) [40]. The programme generally hires a local team to organize and perform the events according to prepared script. The topics to be addressed by folk songs and street theatre are selected by the programme personnel [39]. The folk song and street theatre initiative deliver messages specifically on antenatal care, safe delivery, and postpartum care, family planning, infant and child health [39–43]. From these it could be explained that some of the women who indicated they recall receiving counseling on danger sign in pregnancies from healthcare providers might have also been reinforced by some of these existing interventions in various parts of Bangladesh.

Although some studies have identified a high correlation between age and parity [44, 45], our study found that age and parity do not have the same influence on use of health facility delivery. Whiles health facility delivery increased with age, it decreased with parity. Specifically, the study found that the uptake of health facility delivery increased with age, with older women more likely to utilize health facility delivery compared to younger women. This corroborates other studies in Bangladesh [21] and Ethiopia [35, 46]. The possible explanation to this finding could be that as women grow, they may become more conscious of their health and appreciate the need to deliver in a health facility compared to younger women who might not be well informed about the enormous benefits associated with health facility delivery. Childbearing in advanced age is associated with high risks of complications [47, 48]. It is, therefore, possible that older women in Bangladesh are aware of the risks associated with birth outcomes in advanced maternal age and hence the higher likelihood of health facility delivery.

With parity, the results also showed that women with parity two and above had lower odds to deliver in health facilities. Some literature from Bangladesh [22], Nepal [42], Kenya [24] and Ghana [25] show that women who have more than two births are less likely to deliver in health facilities. As explained by Boah, Mahama and Ayamga [25] and Gebregziabher et al. [49], nulliparous women do not have prior experience with childbirth and due to this may lack self-confidence. Besides, they may consider the delivery experience challenging and for that matter may anticipate unforeseen complications. These might compel them to deliver in a health facility. It is also worthy to acknowledge that first births are at higher risk of experiencing complications that can lead to delivery in a hospital which may also contribute to the higher probability of delivery of first births in health facilities. Nonetheless, a multiparous woman might have gathered enough experience in terms of delivery. If her previous home deliveries were not associated with any complications, this may strengthen her preference for home delivery in the future [25].

We found that there was a positive association between level of education and the odds of delivering in a health facility. Several studies from Bangladesh [20–22] and other LMICs such

as Nepal [50], Cambodia [51], Ethiopia [37, 52, 53], Eretria [49] and India [54] have consistently revealed that high maternal education influences the use of health facility delivery. The link between formal education and the uptake of health facility delivery can be explained through several pathways. First, women who are educated are more likely to comprehend health messages that will be communicated during ANC as well as other ones that are delivered through newspapers, billboards, and other media [22]. Second, formal education erases erroneous beliefs rooted in various traditions that interpret certain healthy behaviors as taboos and by so doing can make educated women acquire positive attitudes towards health facility delivery [22]. Third, as posited by Fekadu, Ambaw and Kidanie [53] and Gebregziabher et al [49] *"higher level of education is associated with better information processing skill, improved cognitive skills and value feelings of self-worth and confidence which all improve health service use."* Also, Tsegay et al. [34] are of the view that formal education is associated with women empowerment and as a result enable women to make better decisions relating to their healthcare and to easily notice complications they might face during pregnancy and delivery. Education increases women's knowledge of where and how the best health care can be accessed and enhances their capability of making autonomous decisions [55]. Fekadu et al [53] also explained that educated women are aware of their health and possess greater knowledge about available health services and the risk of not delivering at a health facility. With this, they are more probable to know and choose the best from the range of services available to them. To this end, it is crucial to invest in female education in Bangladesh to help in the achievement of the Sustainable Development Goals related to maternal and child health [20, 21].

Similar to other studies in Bangladesh [20–22] and other LMICs such as Kenya [56], our results indicated that women having highest wealth quintile had higher odds of health facility delivery. The probable explanation is that poor women may be incapable of paying for some costs associated with health facility delivery including means of getting to the health facility. It is, therefore, necessary to adopt and strengthen existing means of empowering women economically such as the use of microcredit programs which invariable can improve maternal and child health conditions [20, 57].

Our results showed that women in rural areas had lower odds of delivering in a health facility. This corroborates recent studies in Bangladesh [20–22] and other countries [34, 35, 58]. The possible explanations to this finding are that women in the rural areas have varied barriers in accessing health facilities including geographical and challenges in getting information on the need to deliver in health facilities [53]. This is also related to the lower socioeconomic conditions of rural women as well as the paucity of health facilities and personnel in rural Bangladesh [6]. Since more than 65% of the people of Bangladesh live in rural areas [59], there is the need to prioritize those in rural areas to reduce these inequities, which can ultimately help improve maternal, and child health outcomes.

Just like studies from other countries such as Tanzania [33], Uganda [23], and Ethiopia [60], there are variations in the utilization of maternal health services including health facility deliveries in terms of regions or divisions in Bangladesh. We found that women in Khulna had higher odds to deliver in a health facility compared to those in Barisal. Huda et al. [22] explained that in terms of health indicators in Bangladesh, Khulna is one of the best performing divisions. It is therefore imperative for the government of Bangladesh to consider these differences when designing policies related to maternal health to increase health facility deliveries.

## Strengths and limitations

The use of large nationally representative sample makes the study findings generalizable and strengthens the validity of our conclusions. Whereas the findings from this study provide

important considerations for advocacy, some important limitations are worth highlighting. First, the cross-sectional design precludes causality inference but only permits associations between the dependent and independent variable. Another major limitation is that there is also the likelihood of recall bias since the study demanded the women to recall previous event thus recalling whether they have received counseling on pregnancy complications or not. For example, women who experienced danger signs during their pregnancy may be more likely to recall the messages because they were more relevant for them and also more likely to deliver in a facility because of the complications, or women who are more predisposed to use healthcare in general might be more likely to recall the counseling. It is also important to acknowledge that our independent variable is whether women recall receiving counseling on danger signs but not whether they actually received counseling on danger signs. Also, there is the possibility of social desirability bias since. It is worthy to acknowledge also that age and parity are highly correlated in the since young women are more likely to have low parity. Future qualitative studies are needed to explore the in depth information about how women comprehend the information on danger signs in pregnancy during ANC.

## Conclusion

This study has highlighted the importance of telling women about pregnancy complications during ANC. It has revealed that telling women about pregnancy complications during ANC is likely to result in health facility delivery. Covariates such as maternal age, educational level, wealth status, place of residence, division and parity were also associated with health facility delivery. To attain the maternal and child health inclined Sustainable Development Goals, policymakers, program designers and implementers in Bangladesh should consider these results to help reduce maternal and newborn mortality. Specifically, health workers should intensify health education on pregnancy complications during ANC and motivate women to deliver in health facilities.

## Acknowledgments

The authors express their deepest appreciation to Measure DHS for providing them the data for this study.

## Author Contributions

**Conceptualization:** Edward Kwabena Ameyaw, Bright Opoku Ahinkorah, Abdul-Aziz Seidu.

**Data curation:** Edward Kwabena Ameyaw, Bright Opoku Ahinkorah, Abdul-Aziz Seidu.

**Formal analysis:** Edward Kwabena Ameyaw, Bright Opoku Ahinkorah, Abdul-Aziz Seidu.

**Funding acquisition:** Edward Kwabena Ameyaw, Bright Opoku Ahinkorah, Abdul-Aziz Seidu.

**Investigation:** Edward Kwabena Ameyaw, Bright Opoku Ahinkorah, Abdul-Aziz Seidu.

**Methodology:** Edward Kwabena Ameyaw, Bright Opoku Ahinkorah, Abdul-Aziz Seidu.

**Project administration:** Edward Kwabena Ameyaw, Bright Opoku Ahinkorah, Abdul-Aziz Seidu.

**Resources:** Edward Kwabena Ameyaw, Bright Opoku Ahinkorah, Abdul-Aziz Seidu.

**Software:** Edward Kwabena Ameyaw, Bright Opoku Ahinkorah, Abdul-Aziz Seidu.

**Supervision:** Edward Kwabena Ameyaw, Bright Opoku Ahinkorah, Abdul-Aziz Seidu.

**Validation:** Edward Kwabena Ameyaw, Bright Opoku Ahinkorah, Abdul-Aziz Seidu.

**Visualization:** Edward Kwabena Ameyaw, Bright Opoku Ahinkorah, Abdul-Aziz Seidu.

**Writing – original draft:** Edward Kwabena Ameyaw, Bright Opoku Ahinkorah, Abdul-Aziz Seidu.

**Writing – review & editing:** Edward Kwabena Ameyaw, Bright Opoku Ahinkorah, Abdul-Aziz Seidu.

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
