## [Decision Letter · Decision Letter 0]

17 Dec 2019

PONE-D-19-25087

Does knowledge of pregnancy complications influence health facility delivery? Analysis of 2014 Bangladesh Demographic and Health Survey

PLOS ONE

Dear Mr Seidu,

Thank you for submitting your manuscript to PLOS ONE. After careful consideration, we feel that it has merit but does not fully meet PLOS ONE’s publication criteria as it currently stands. Therefore, we invite you to submit a revised version of the manuscript that addresses the points raised during the review process.

We would appreciate receiving your revised manuscript by Jan 31 2020 11:59PM. To enhance the reproducibility of your results, we recommend that if applicable you deposit your laboratory protocols in protocols.io, where a protocol can be assigned its own identifier (DOI) such that it can be cited independently in the future. For instructions see: http://journals.plos.org/plosone/s/submission-guidelines#loc-laboratory-protocols

We look forward to receiving your revised manuscript.

Kind regards,

Michael Johnson Mahande, PhD

Academic Editor

PLOS ONE

Journal Requirements:

**When submitting your revision, we need you to address these additional requirements:**

**Please ensure that your manuscript meets PLOS ONE's style requirements, including those for file naming. The PLOS ONE style templates can be found at http://www.plosone.org/attachments/PLOSOne_formatting_sample_main_body.pdf and http://www.plosone.org/attachments/PLOSOne_formatting_sample_title_authors_affiliations.pdf**

Additional Editor Comments (if provided):

Major revision

Reviewers' comments:

Reviewer's Responses to Questions

**Comments to the Author**

1. Is the manuscript technically sound, and do the data support the conclusions?

Reviewer #1: Partly

2. Has the statistical analysis been performed appropriately and rigorously? 

Reviewer #1: Yes

3. Have the authors made all data underlying the findings in their manuscript fully available?

Reviewer #1: Yes

4. Is the manuscript presented in an intelligible fashion and written in standard English?

Reviewer #1: No

5. Review Comments to the Author

Reviewer #1: Counseling on danger signs that require medical attention during pregnancy and delivery is an important part of antenatal care and is actively promoted by maternal health interventions in Bangladesh. The analysis in this paper examines the association between counseling on complications during ANC and delivery in a health facility in Bangladesh. Below are some comments to the authors.

Statistical Issues

1. The sample for the statistical analysis is not adequately described in the Methods section (p. 6). In the 2014 Bangladesh DHS women who had a live birth in the 3 years (not in the last 5 years) before the survey were asked about antenatal care for their most recent live birth. Only those women who had at least one ANC visit with any provider were asked about counseling on danger signs during pregnancy. Therefore your analysis sample is women who had at least one live birth in the 3 years before the survey and had at least one ANC visit.

2. There is some selection into this sample that is not discussed in that women who did not have ANC are likely to be different from those who did. Therefore your results are not representative of all women who had a live birth in the 3 years before the survey. I recommend stating the percentage of women with a recent live birth who had at least one ANC visit so the reader can see how many women are excluded.

3. It is important to be clear that what you are measuring with the independent variable is whether women recall receiving counseling on danger signs not whether they actually received counseling on danger signs. The difference is subtle, in that women have to recall the messages in order to act on them so recall is what is important for action, but the recommendations are different if women were counseled but don’t recall the message rather than that they weren’t counseled at all. You can’t make that distinction with these data which should be addressed in the Discussion.

4. There is also selection in who recalls being counseled on danger signs. For example, women who experienced danger signs may be more likely to recall the messages because they were more relevant for them and also more likely to deliver in a facility because of the complications, or women who are more predisposed to use healthcare in general might be more likely to recall the counseling. This kind of selection will bias your results and limits causal interpretation. This is difficult to address statistically (instrumental variables are used to address similar recall selection issues in health communication program evaluation for example, but finding good instruments can be very difficult if not impossible and weak instruments will introduce different types of biases). You mention recall bias very briefly at the end of the Strengths and Limitations section on p. 17 but this issue is a significant limitation and warrants more attention than currently given. I recommend also being more cautious in the causal interpretation of your findings as what you estimate is association rather than causal association given these selection issues.

5. Age and parity are highly correlated (young women are more likely to be low parity). Your discussion of the results for these two variables in the Discussion should acknowledge this. There are only 9 births (weighted) to women age 45-49 so your 95% CI for the AOR for this category in Table 2 is very wide. I suggest combining the 40-44 and 45-49 age groups in the analysis.

Other comments

6. The description of the DHS survey on pages 5-6 needs tightening up. Mostly this is editorial but the last sentence of the Data Source section (lines 112-113) is misleading. The number of households identified is not equal to the number of eligible women identified as implied by this sentence (i.e. not every household will contain an eligible woman and some households will contain more than 1 eligible woman). In the 2014 BDHS, 17,989 households were selected for the survey of which 17,300 were successfully interviewed. In those households 18,245 women eligible for the individual interview were identified, of which 17,863 were interviewed. Providing more complete information like this allows the reader to also get a sense of the response rates in the survey, which is important for assessing quality.

7. In Table 1 you don’t really need to include both the numerator (n) and the % for the place of delivery variables. The % is the most useful information for the reader but that is in parentheses so is less prominent. It would be clearer to only present the %. Also, is the n presented weighted or unweighted in the place of delivery columns? If you divide the n in the place of delivery column by the n for the category you don’t get the percentage reported which I assume is related to weighting (e.g. for those not told about danger signs during ANC, the weighted denominator is 1,875 and the numerator given for delivering at home is 1,046 so the % implied is 55.8% not 56.4%. I think 56.4% is correct though not 55.8% because the 2014 BDHS report reports 56.9% of women who received ANC received counseling on danger signs).

8. The focus of the paper is on the association between counseling on pregnancy danger signs during ANC and delivery care but the Discussion spends a lot of time discussing the results for the control variables which are not the focus of the paper (although it relates them nicely to other literature). This is particularly the case for education. The Discussion would benefit from being more focused on the primary independent variable of interest while acknowledging that associations between control variables and delivery care are largely as expected. Also, in the discussion of the parity results be aware that that first births are at higher risk of experiencing complications that can lead to delivery in a hospital which may also contribute to the higher probability of delivery of first births in health facilities (the DHS data don’t allow you to distinguish whether a woman chose a priori to deliver in a hospital or started delivering at home and then had to be taken to a facility due to complications).

9. I would like to see the discussion place the results in the existing program context in Bangladesh more. Bangladesh has interventions to promote counseling on danger signs during ANC including behavior change communication posters and counseling guides, although fewer than 50% of women in this analysis recalled receiving those messages. And it already considers divisional variation in its health programming, although disparities remain. The current discussion does not demonstrate knowledge of the Bangladesh program environment. There is also a significant shift going on towards accessing maternity care in the private sector which has implications for who and how to target interventions to strengthen counseling on danger signs. And there is a significant increase in caesarean sections associated with increased delivery in private sector facilities. Some integration of these larger contextual factors into the Discussion would be beneficial. There are also some new results coming out now from the 2017 Bangladesh DHS and 2016 Bangladesh maternal mortality and morbidity survey that would be useful to integrate into the discussion (although I recognize that the preliminary 2017 DHS results were not available when you wrote this paper).

10. The paper overall would benefit from review by a technical editor to tighten up and clarify the writing. Several sentences are unclear or imprecise. For example lines 68-70 (awkward sentence), 131-133 (unclear), lines 145-146 (imprecise as you are controlling for confounding factors not fitting interactions), 150-151 ( “the study” here refers to the DHS not the current study you are presenting in the paper but that is not immediately clear) etc.

6. PLOS authors have the option to publish the peer review history of their article (what does this mean?). If published, this will include your full peer review and any attached files.

Reviewer #1: Yes: Sian L Curtis

---

## [Author Response · Author response to Decision Letter 0]

13 Jan 2020

AUTHOR’S RESPONSE TO REVIEWS

Title: Does knowledge of pregnancy complications influence health facility delivery? Analysis of 2014 Bangladesh Demographic and Health Survey 

Ref: PONE-D-19-25087

Version: 1

Date: 11/01/2020

Dear Editor/Reviewer (s),

Thank you for the opportunity to revise our manuscript entitled “Does knowledge of pregnancy complications influence health facility delivery? Analysis of 2014 Bangladesh Demographic and Health Survey”. Please we have considered all the comments and modified our manuscript in that respect. In the following detailed response, we address each comment calling for changes point-by-point, indicating where relevant additional texts have been added to the body of the manuscript. Most of the changes have been indicated in Yellow. We believe the manuscript has improved substantively and will be published in your reputable journal. 

 Review Comments to the Author

Reviewer #1: Counseling on danger signs that require medical attention during pregnancy and delivery is an important part of antenatal care and is actively promoted by maternal health interventions in Bangladesh. The analysis in this paper examines the association between counseling on complications during ANC and delivery in a health facility in Bangladesh. Below are some comments to the authors.

Statistical Issues

Comment: 1. The sample for the statistical analysis is not adequately described in the Methods section (p. 6). In the 2014 Bangladesh DHS women who had a live birth in the 3 years (not in the last 5 years) before the survey were asked about antenatal care for their most recent live birth. Only those women who had at least one ANC visit with any provider were asked about counseling on danger signs during pregnancy. Therefore, your analysis sample is women who had at least one live birth in the 3 years before the survey and had at least one ANC visit.

Response: Please we have described this in detail at the methods section of the manuscript (see Page 6, line 128-137). Also, the report clearly states 3 years but not 5.

Comment: 2. There is some selection into this sample that is not discussed in that women who did not have ANC are likely to be different from those who did. Therefore, your results are not representative of all women who had a live birth in the 3 years before the survey. I recommend stating the percentage of women with a recent live birth who had at least one ANC visit so the reader can see how many women are excluded.

Response: Please we have stated the number of women who were excluded in the methods section of the manuscript (Page 6, line 136-137).

Comment: 3. It is important to be clear that what you are measuring with the independent variable is whether women recall receiving counseling on danger signs not whether they actually received counseling on danger signs. The difference is subtle, in that women have to recall the messages in order to act on them so recall is what is important for action, but the recommendations are different if women were counseled but don’t recall the message rather than that they weren’t counseled at all. You can’t make that distinction with these data which should be addressed in the Discussion.

Response: Many thanks for this clarification. Please we have acknowledged this as a limitation in the discussion section of our manuscript (Page 17, line 331-347). 

Comment: 4. There is also selection in who recalls being counseled on danger signs. For example, women who experienced danger signs may be more likely to recall the messages because they were more relevant for them and also more likely to deliver in a facility because of the complications, or women who are more predisposed to use healthcare in general might be more likely to recall the counseling. This kind of selection will bias your results and limits causal interpretation. This is difficult to address statistically (instrumental variables are used to address similar recall selection issues in health communication program evaluation for example, but finding good instruments can be very difficult if not impossible and weak instruments will introduce different types of biases). You mention recall bias very briefly at the end of the Strengths and Limitations section on p. 17 but this issue is a significant limitation and warrants more attention than currently given. I recommend also being more cautious in the causal interpretation of your findings as what you estimate is association rather than causal association given these selection issues.

Response: We are much grateful for this comment. Please we have discussed recall bias in detail in the limitation section of our manuscript (Page 17-18, line 331-347). 

Comment: 5. Age and parity are highly correlated (young women are more likely to be low parity). Your discussion of the results for these two variables in the Discussion should acknowledge this. 

Response: We acknowledge this and have added this to the discussion and as a limitation (Page 14 line 260) and (Page 17 line 343-347). 

Comment: There are only 9 births (weighted) to women age 45-49 so your 95% CI for the AOR for this category in Table 2 is very wide. I suggest combining the 40-44 and 45-49 age groups in the analysis. 

Response: Please we have recategorised age by combining 40-44 and 45-49 age groups in the analysis as well as religious affiliation (Please Table 1). 

Other comments

Comment: 6. The description of the DHS survey on pages 5-6 needs tightening up. Mostly this is editorial but the last sentence of the Data Source section (lines 112-113) is misleading. The number of households identified is not equal to the number of eligible women identified as implied by this sentence (i.e. not every household will contain an eligible woman and some households will contain more than 1 eligible woman). In the 2014 BDHS, 17,989 households were selected for the survey of which 17,300 were successfully interviewed. In those households 18,245 women eligible for the individual interview were identified, of which 17,863 were interviewed. Providing more complete information like this allows the reader to also get a sense of the response rates in the survey, which is important for assessing quality.

Response: We appreciate this a lot! Please we have revised this section (see Page 5, line 110-113). 

Comment: 7. In Table 1 you don’t really need to include both the numerator (n) and the % for the place of delivery variables. The % is the most useful information for the reader but that is in parentheses so is less prominent. It would be clearer to only present the %. Also, is the n presented weighted or unweighted in the place of delivery columns? If you divide the n in the place of delivery column by the n for the category you don’t get the percentage reported which I assume is related to weighting (e.g. for those not told about danger signs during ANC, the weighted denominator is 1,875 and the numerator given for delivering at home is 1,046 so the % implied is 55.8% not 56.4%. I think 56.4% is correct though not 55.8% because the 2014 BDHS report reports 56.9% of women who received ANC received counseling on danger signs).

Response: We have presented only the percentages (See Table 1). We acknowledge the percentages might be a bit different due to weighting and also due to the fact that we omitted those who indicated, “don’t know”(4, [0.11%]) from the sample. 

Comment: 8. The focus of the paper is on the association between counseling on pregnancy danger signs during ANC and delivery care but the Discussion spends a lot of time discussing the results for the control variables which are not the focus of the paper (although it relates them nicely to other literature). This is particularly the case for education. The Discussion would benefit from being more focused on the primary independent variable of interest while acknowledging that associations between control variables and delivery care are largely as expected. Also, in the discussion of the parity results be aware that that first births are at higher risk of experiencing complications that can lead to delivery in a hospital which may also contribute to the higher probability of delivery of first births in health facilities (the DHS data don’t allow you to distinguish whether a woman chose a priori to deliver in a hospital or started delivering at home and then had to be taken to a facility due to complications).

Response: We really appreciate this comment and have modified the discussion section of the manuscript (Page 11-17).

Comment: 9. I would like to see the discussion place the results in the existing program context in Bangladesh more. Bangladesh has interventions to promote counseling on danger signs during ANC including behavior change communication posters and counseling guides, although fewer than 50% of women in this analysis recalled receiving those messages. And it already considers divisional variation in its health programming, although disparities remain. The current discussion does not demonstrate knowledge of the Bangladesh program environment. There is also a significant shift going on towards accessing maternity care in the private sector which has implications for who and how to target interventions to strengthen counseling on danger signs. And there is a significant increase in caesarean sections associated with increased delivery in private sector facilities. Some integration of these larger contextual factors into the Discussion would be beneficial. There are also some new results coming out now from the 2017 Bangladesh DHS and 2016 Bangladesh maternal mortality and morbidity survey that would be useful to integrate into the discussion (although I recognize that the preliminary 2017 DHS results were not available when you wrote this paper).

Response: We appreciate this comment. Please we have integrated these in our discussion (Page 11-14). 

Comment: 10. The paper overall would benefit from review by a technical editor to tighten up and clarify the writing. Several sentences are unclear or imprecise. For example lines 68-70 (awkward sentence), 131-133 (unclear), lines 145-146 (imprecise as you are controlling for confounding factors not fitting interactions), 150-151 ( “the study” here refers to the DHS not the current study you are presenting in the paper but that is not immediately clear) etc.

Response: We have proof read the entire manuscript to correct these errors. 

Thank you once again for meticulously reviewing this manuscript and giving us insightful comments to improve upon this manuscript.

---

## [Editor Report · Decision Letter 1]

7 Aug 2020

Does knowledge of pregnancy complications influence health facility delivery? Analysis of 2014 Bangladesh Demographic and Health Survey

PONE-D-19-25087R1

Dear Dr. Abdul-Aziz Seidu,

We’re pleased to inform you that your manuscript has been judged scientifically suitable for publication and will be formally accepted for publication once it meets all outstanding technical requirements.

Kind regards,

Georg M. Schmölzer

Academic Editor

PLOS ONE
---

## [Editor Report · Acceptance letter]

18 Aug 2020

PONE-D-19-25087R1 

Does knowledge of pregnancy complications influence health facility delivery? Analysis of 2014 Bangladesh Demographic and Health Survey 

Dear Dr. Seidu:

I'm pleased to inform you that your manuscript has been deemed suitable for publication in PLOS ONE. Congratulations! Your manuscript is now with our production department. 

Kind regards, 

on behalf of

Dr. Georg M. Schmölzer 

Academic Editor

PLOS ONE